# Colistin plasma concentrations are not associated with better clinical outcomes in patients with pneumonia caused by extremely drug-resistant *Pseudomonas aeruginosa*

Luisa Sorlí,[1,2,3,4] Sonia Luque,[4,5,6] Jian Li,[7] Adela Benítez-Cano,[8] Xenia Fernández,[7] Nuria Prim,[9] Victoria Vega,[10] Joan Gómez-Junyent,[1,3] Inmaculada López-Montesinos,[1] Silvia Gómez-Zorrilla,[1,3,4] M. Milagro Montero,[1,3,4] Santiago Grau,[3,4,7] Juan Pablo Horcajada[1,3,4]

**ABSTRACT**   The objective of this study was to examine the relationship between colistin plasma concentrations and clinical outcomes in patients with hospital-acquired pneumonia (HAP) caused by extensively drug-resistant *Pseudomonas aeruginosa* (XDR-PA). A prospective observational cohort study was conducted at a tertiary care hospital. Patients diagnosed with HAP caused by XDR-PA between January 2010 and September 2018 were included. Steady-state plasma concentrations ($C_{ss,avg}$) of colistin were measured by high performance liquid chromatography (HPLC). Based on the pharmacokinetic/pharmacodynamic data of colistin in animal models, previous clinical data, and the minimum inhibitory concentration (MIC) of the most prevalent strain in our center, an area under the ROC curve (AUC)/$MIC_{24h}$ between 30 and 60 mg·h/L was considered optimal. The primary outcome was 30-day mortality, and the secondary outcomes were clinical cure and colistin-associated nephrotoxicity. Seventy-five patients were included. The median $C_{ss,avg}$ was 1.1 (0.56–1.75) mg/L and only 23 (30.7%) patients achieved an AUC/$MIC_{24h}$ of 30–60 mg·h/L and the 30-day mortality was 30.7%. In multivariate analysis, sequential organ failure assessment score [adjusted hazard ratio (95% confidence interval, CI) 1.19 (1.07–1.32)] and high AUC/$MIC_{24h}$ (hazard ratio 1.55, 95% CI 1.17–2.04, $P = 0.002$) were associated with increased risk of death and clinical failure. Based on the maximally selected standardized log-rank statistic, the $C_{ss,avg}$ cutoff point that best predicted mortality was 1.5 mg/L. Nephrotoxicity at the end of treatment was 38.7%. High colistin exposure was not associated with improved clinical outcomes in the setting of HAP caused by XDR-PA. These data suggest that caution is required with intravenous colistin for the treatment of HAP caused by XDR-PA.

**IMPORTANCE**   In some cases, colistin is the only treatment option for infections caused by the very drug-resistant *Pseudomonas aeruginosa*. However, in the past decade, there have been questions concerning its pharmacokinetics and concentration at the site of infection. In this scenario, its use in a difficult-to-treat infection like pneumonia is currently debatable. This is a clinical pharmacokinetic study of colistin in patients with multidrug-resistant P. aeruginosa pneumonia. Our findings demonstrate that colistin exposure is associated with worse clinical outcomes rather than better clinical outcomes, implying that other therapeutic options should be explored in this clinical setting.

**KEYWORDS**   *Pseudomonas aeruginosa*, pneumonia, colistin, pharmacokinetics

Address correspondence to Luisa Sorlí, Lsorli@psmar.cat.

Luisa Sorlí and Sonia Luque contributed equally to this article. The clinical profile of the publication was taken into consideration to determine author order.

The authors declare no conflict of interest.

See the funding table on p. 10.

Hospital-acquired pneumonia (HAP) is a common healthcare-acquired infection worldwide. It includes two distinct subgroups: non-ventilator HAP (NV-HAP) and ventilator-associated pneumonia (VAP). Both types of infection are among the most

common in the hospital setting and have placed a significant burden on the health-care system. *Pseudomonas aeruginosa* is among the most common causes of HAP worldwide, with a high proportion of multidrug resistance (MDR) (1, 2). A recent study in Greece, Italy, and Spain found that 35.8% of P. aeruginosa isolates from people with VAP were extensively drug-resistant (XDR), 30.2% were multidrug-resistant (MDR), and 3.2% were pandrug-resistant (PDR) (3).

The number of studies that have specifically assessed different treatment options for the treatment of XDR *P. aeruginosa* pneumonia is limited. Both ceftazidime-avibactam and ceftolozane-tazobactam were recently shown to be non-inferior to meropenem for the treatment of HAP, including those caused by *P. aeruginosa* (4, 5). However, resistance has been reported with these agents (6–8) and sometimes the only available treatment is colistin. Recently, new data was released on the pharmacokinetics (PK) of colistin and its prodrug colistimethate sodium (CMS) in different patient groups. However, the PK-clinical efficacy relationship is still not fully understood (910 ); PK/PD (pharmacodynamic) data from mouse lung infection models suggests that the current PK/PD targets may not be right for pneumonia (11). Furthermore, there are few data on the effectiveness of intravenous CMS in patients with XDR P. aeruginosa pneumonia. The aims of this study, therefore, were to examine the relationship between colistin plasma concentrations and clinical outcomes in patients diagnosed with HAP caused by XDR *P. aeruginosa* and to assess the safety of colistin in this clinical scenario.

## RESULTS

Seventy-five patients diagnosed with HAP caused by XDR *P. aeruginosa* and treated with intravenous CMS were included. Demographic characteristics, disease severity, and treatment data are summarized in Table 1. Of note, the median plasma concentration of formed colistin was 1.1 (0.56–1.75) mg/L and only 30.7% of patients (23/75) achieved an $AUC_{24h}$/MIC (minimum inhibitory concentration) of 30–60 mg·h/L or $C_{ss,avg}$ of 0.625–1.125 mg/L. The distribution of dose-normalized colistin plasma concentrations in relation to baseline estimated glomerular filtration rate (eGFR) is shown in Fig. S1.

### Treatment outcomes

Overall, 30-day mortality was 30.7%. Risk factors associated with mortality and clinical failure in univariate analysis are shown in Table 2. No differences were observed for any of the treatment characteristics (dosage and therapeutic regimen, use of monotherapy or combined therapy, use of nebulized colistin), except that the patients who died received shorter courses of treatment. With respect to the PK data, patients who died had a higher $AUC_{24h}$/MIC ratio and were more commonly in the overexposure group ($C_{ss,avg} \geq 1.25$ mg/L). Although nephrotoxicity was also more frequent in patients who died, this difference was not statistically significant.

Multivariate analysis for time to death is shown in Table 3. This analysis found two significant predictors: (i) SOFA score [hazard ratio (HR) 1.19, 95% confidence interval (CI) 1.07–1.32, $P = 0.002$] and (ii) $C_{ss, avg}$ (HR 1.55, 95% CI 1.17–2.04, $P = 0.002$). $C_{ss}$ was also evaluated according to the degree of exposure. The multivariate analysis performed with this approach is shown in Table S3. Of note is the high risk of mortality in patients in the overexposure group (HR 4.57, 95% CI 1.46–14.3, $P = 0.009$). Figure 1 shows the Kaplan-Meier curves for 30-day mortality according to the predefined target colistin plasma concentration of 2mg/L. Figure S2 shows the Kaplan-Meier curves for mortality based on colistin plasma concentration stratified by degree of exposure. Both graphs illustrate the relationship between higher colistin plasma concentrations and time to death. We then attempted to discover the cutoff at which patients had a dismal prognosis. Based on the maximally selected standardized log-rank statistic, the $C_{ss,avg}$ cutoff point that best predicted mortality was 1.5 mg/L (Fig. S3). With respect to clinical cure, multivariate analysis showed similar results. Factors independently related to clinical failure were SOFA score (OR 1.36, 95% CI 1.1–1.67, $P = 0.004$) and $C_{ss,avg}$ (OR 1.80, 95% CI 1.05–3.10, $P = 0.032$). When the same analysis was performed with $C_{ss,avg}$ by degree of exposure,

**TABLE 1** Characteristics of 75 patients with HAP caused by XDR *Pseudomonas aeruginosa* treated with colistin[b]

| Characteristics | Value[a] |
|---|---|
| Demographic parameters | |
| Age (years) | 71.6 (61.2–77.0) |
| Sex (male) | 59 (78.7) |
| Concomitant diseases | |
| Chronic pulmonary disease | 41 (54.7) |
| Solid tumor | 25 (33.3) |
| Cardiac disease | 24 (32.0) |
| Diabetes mellitus | 19 (25.3) |
| Chronic kidney disease | 18 (24) |
| Leukemia and lymphoma | 7 (9.33) |
| Liver disease | 6 (8.0) |
| Charlson comorbidity index | 4.9 ± 2.5 |
| Clinical condition | |
| ICU admission | 28 (37.3) |
| SOFA score | 2 (1-6)[c] |
| eGFR at baseline (mL/min/1.73m2) | 92.5 (51.6–168.4) |
| BMI (kg/m2) | 24.4 (22.0–28.9) |
| NV-HAP | 68 (90.7) |
| VAP | 7 (9.3) |
| CMS treatment | |
| Duration (days) | 14 (10–21) |
| CMS daily dose (million IU/day) | 5.54 ± 2.36 |
| CMS daily dose (mg/kg/day) | 6.51 ± 3.21 |
| CBA daily dose (mg/day) | 189.5 ± 80.2 |
| CBA daily dose (mg/kg/day) | 2.77 ± 1.36 |
| Loading dose | 6 (10.7) |
| Route of CMS administration | 32 (42.7) |
| Concurrent use of IV/inhaled colistin | 43 (57.3) |
| Inhaled CMS dose (million IU) | 3 (1-6) |
| Combination therapy (>3 days) | 42 (56.0) |
| With carbapenems | 6 (8) |
| With aminoglycosides | 9 (12) |
| With cephalosporins | 22 (29.3) |
| Pharmacokinetic data | |
| $C_{ss,\ avg}$ (mg/L) | 1.1 (0.56–1.75) |
| $C_{ss,\ avg}$ < 0.625 mg/L | 21 (28) |
| $C_{ss,\ avg}$ 0.625–1.25 mg/L | 23 (30.7) |
| $C_{ss,\ avg}$ > 1.25 mg/L | 31 (41.3) |
| $C_{ss,\ avg}$ > 2 mg/L | 13 (17.3) |
| $AUC_{24h}$/MIC | 43.3 (21.6–70.4) |
| Colistin-associated nephrotoxicity | 29 (38.7) |
| R | 8/29 (27.6) |
| I | 18/29 (62.1) |
| F | 3/29(10.3) |
| Microbiological outcomes | |
| Eradication | 26 (34.7) |
| Colonization | 20 (26.7) |
| No follow-up | 25 (33.3) |
| Superinfection | 4 (5.33) |
| Clinical outcomes | |
| Clinical response of pneumonia | 51 (68.0) |

*(Continued on next page)*

**TABLE 1** Characteristics of 75 patients with HAP caused by XDR *Pseudomonas aeruginosa* treated with colistin[b] (*Continued*)

| Characteristics | Value[a] |
|---|---|
| 30-day all-cause mortality | 23 (30.7) |

[a]Categorical data are expressed as *n* (%), and continuous data are expressed as mean ± standard deviation or median (interquartile range).
[b]ICU, intensive care unit; SOFA, Sequential Organ Failure Assessment score; eGFR, estimated glomerular filtration rate; BMI, body mass index; NV-HAP, non-ventilator associated hospital-acquired pneumonia; VAP, ventilator-associated pneumonia; CMS, colistimethate sodium; IU, international units; CBA, colistin base activity; IV, intravenous; IH, inhaled; $C_{ss,avg}$, colistin plasma concentration at steady state.
[c]SOFA score 2 (1–6) are the values of interquartile range (p25-p75). m2 represents square metre. Inhaled CMS in million of IU: (1-6) represents Interquartile range. Comibnation therapy with carbapenems 6 patients (8%) and with aminoglycosides 9 patients (12%).

there was a non-significant trend toward a higher degree of clinical cure in patients with overexposure (Table S4).

Colistin-associated nephrotoxicity at the end of treatment was 38.7%. However, most patients showed mild nephrotoxicity according to the RIFLE [Risk (GFR decrease > 25%), Injury (GFR decrease > 50%), Failure (GFR decrease > 75%), Loss (complete loss of kidney function > 4 weeks), and End-stage kidney disease (End-stage kidney disease > 3 months)] criteria (27.6% R, 62.1% I, and only 10.3% F). Colistin plasma concentrations were higher in patients with nephrotoxicity at the end of treatment (1.5 vs 0.9 mg/L, *P* = 0.001), and patients with colistin-associated nephrotoxicity had a worse clinical outcome, although this association was not statistically significant.

**TABLE 2** Univariate analysis of factors associated with clinical failure and 30-day all-cause mortality[a]

| Variables | Clinical success<br>N = 51 | Clinical failure<br>N = 24 | P value | Survived<br>N = 52 | Died<br>N = 23 | P value |
|---|---|---|---|---|---|---|
| Demographic parameters | | | | | | |
| Age (years) | 67.6 (54.8–76.6) | 72.8 (65.1–79.3) | 0.17 | 68.3 (55.1–75.8) | 72.6 (64.9–80.4) | 0.15 |
| Sex (male) | 39 (76.5) | 20 (83.3) | 0.50 | 39 (75) | 20 (87.0) | 0.24 |
| Charlson comorbidity index | 4.41 ± 2.54 | 5.83 ± 2.28 | **0.02[b]** | 4.44 ± 2.55 | 5.83 ± 2.23 | **0.028** |
| Clinical condition | | | | | | |
| ICU admission | 14 (27.5) | 14 (58.3) | **0.01** | 15 (28.9) | 13 (56.5) | **0.022** |
| SOFA | 2 (1–4) | 6 (2–8.5) | **0.0009** | 2 (1–4) | 6 (2–9) | **0.01** |
| eGFR (mL/min/1.73 m²) at baseline | 103.1 (69.5–199.7) | 72.7 (40.6–106.4) | **0.0065** | 101.2 (60.3–194.5) | 75.4 (48.2–124.8) | **0.062** |
| BMI (kg/m²) | 24.7 (21.8–28.9) | 23.9 (22.2–28.3) | 0.80 | 24.7 (2–28.9) | 23.5 (22.0–28.7) | 0.80 |
| Treatment | | | | | | |
| Duration (days) | 16 (12–22) | 10 (8–15.5) | **0.0017** | 16 (12–22) | 10 (8–16) | **0.006** |
| CMS daily dose (million IU/day) | 5.67 ± 2.29 | 5.27 ± 2.53 | 0.50 | 5.42 ± 2.39 | 5.80 ± 2.32 | 0.52 |
| CMS daily dose (mg/kg/day) | 6.63 ± 3.32 | 6.24 ± 3.0 | 0.64 | 6.35 ± 3.40 | 6.88 ± 2.72 | 0.52 |
| CBA daily dose (mg/day) | 192.7 ± 78.0 | 182.6 ± 86.4 | 0.62 | 184.4 ± 81.5 | 201.7 ± 77.9 | 0.40 |
| CBA daily dose (mg/kg/days | 2.81 ± 1.41 | 2.65 ± 1.27 | 0.64 | 2.70 ± 1.45 | 2.92 ± 1.16 | 0.52 |
| Combination therapy (>3 days) | 36 (70.6) | 17 (70.8) | 1 | 36 (69.2) | 17 (73.9) | 0.68 |
| Nebulized colistin | 28 (54.9) | 15 (62.5) | 0.535 | 27 (62.8) | 16 (69.6) | 0.15 |
| Pharmacokinetic data | | | | | | |
| $C_{ss,avg}$ (mg/L) | 1.08 ± 1.0 | 1.89 ± 1.12 | **0.0024** | 1.03 ± 0.93 | 2.03 ± 1.17 | **0.0002** |
| $C_{ss, avg}$ optimal (0.625–1.25) | 17 (33.3) | 6 (25) | **0.024** | 19 (36.5) | 4 (17.4) | **0.004** |
| $C_{ss, avg}$ subtherapeutic (<0.625) | 18 (35.3) | 3 (12.5) | | 18 (34.6) | 3(13) | |
| $C_{ss, avg}$ supratherapeutic (≥1.25) | 16 (31.4) | 15 (62.5) | | 16 (30.8) | 15 (65.2) | |
| $C_{ss, avg}$ ≥ 2 mg/L | 4 (7.84) | 9 (37.5) | **0.003** | 3 (5.77) | 10 (43.5) | **<0.001** |
| AUC/MIC | 44.6 ± 42.6 | 75.0 ± 46.6 | **0.006** | 42.0 ± 38.0 | 82.2 ± 50.44 | **0.0003** |
| Colistin- associated nephrotoxicity | 18 (35.3) | 11 (45.8) | 0.38 | 18 (34.6) | 11 (47.8) | 0.28 |

[a]ICU, intensive care unit; SOFA, Sequential Organ Failure Assessment score; eGFR, estimated glomerular filtration rate; BMI, body mass index; NN, nosocomial pneumonia; VAP, ventilator-associated pneumonia; CMS, colistimethate sodium; IU, international units; CBA, colistin base activity; IV, intravenous; IH, inhaled; $C_{ss, avg}$, colistin plasma concentration at steady state.
[b]Boldface indicates values with statistical significance in univariate analysis.

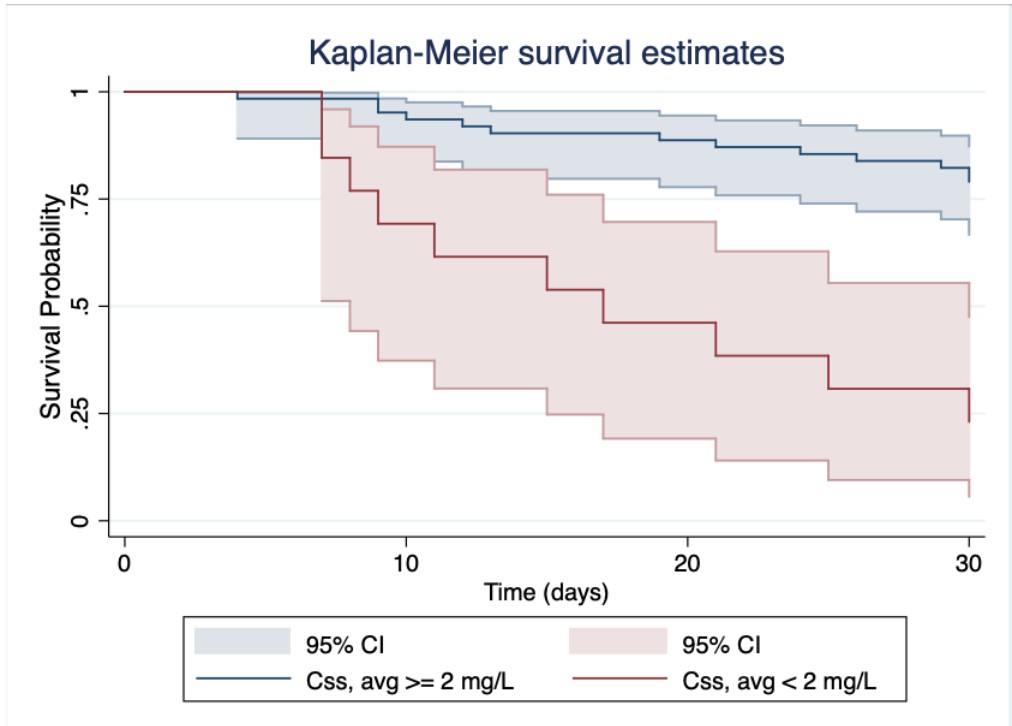

**FIG 1** Cumulative Kaplan-Meier estimates of the probability of overall 30-day survival. The estimates are stratified based on colistin plasma concentrations at steady-state, with a cutoff value of 2 mg/L for $C_{ss,avg}$. Significant differences in 30-day survival between patients with $C_{ss,avg}$ above or below 2 mg/L.

**TABLE 3** Multivariate analysis of risk factors associated with 30-day all-cause mortality ($C_{ss, avg}$ as continuous variable)

| Variable | HR | P | 95% CI |
|---|---|---|---|
| SOFA | 1.18 | 0.002 | 1.07–1.32 |
| $C_{ss, avg}$ (mg/L) | 1.54 | 0.002 | 1.17–2.04 |
| Charlson comorbidity index | 1.04 | 0.531 | 0.91–1.20 |

Finally, a positive microbiological result was documented in 26 (34.7%) patients, and colonization in 20 (26.7%). Unfortunately, 25 (33.3%) patients had no follow-up cultures. There was no correlation between colistin plasma concentrations and microbiological outcomes ($P = 0.447$) or between inhaled colistin use and microbiological outcome ($P = 0.664$).

## DISCUSSION

This study was conducted over 8 years and included 75 patients with HAP caused by XDR *P. aeruginosa* and treated with CMS. To our knowledge, this is the largest patient cohort studied to date regarding the impact of colistin plasma concentrations in this difficult-to-treat infection. Surprisingly, our findings showed that higher plasma exposure to colistin was not associated with better clinical outcomes, especially when colistin plasma concentrations were higher than 1.5 mg/L, which is a lower value than the predefined $C_{ss,avg}$ target of 2 mg/L.

A high mortality rate (30.7%) was observed in this series. Previous studies have evaluated the use of intravenous CMS in patients with infections caused by XDR *P. aeruginosa*. In a recent study of patients diagnosed with HAP caused by XDR *P. aeruginosa*, Khawcharoenporn et al. reported a survival rate of 51% in patients treated with active monotherapy and 90% in those treated with two-drug combination therapy

(carbapenems, fosfomycin, and piperacillin-tazobactam), which compares with 72.7% clinical cure in patients with active monotherapy vs 67.9% in patients with combination therapy in our study. Nevertheless, the findings of the two studies cannot be easily compared due to differences in several key aspects. First, the patients included in the study by Khawcharoenporn et al. were more severely ill than those included in the present one [APACHE II at the onset of XDR *P. aeruginosa* pneumonia was 16 (interquartile range, IQR 12–23) vs 14 (IQR 9–20)]. Second, and perhaps more importantly, in our study, only 10.7% of patients received a loading dose, whereas Khawcharoenporn et al. administered a loading dose of 300 mg colistin. Furthermore, they administered the accompanying antibiotics in continuous infusion, which increased the likelihood that the PK/PD target would be achieved. This may explain the higher rate of clinical cure in the group that received combination therapy in the study cited above (12). Levin et al. reported 20 patients with pneumonia: 6 caused by *P. aeruginosa* and 14 by *Acinetobacter baumannii*. Patients included in that study were younger but had similar APACHE II scores to those included in the present study (13.1 ± 7.0 vs 14.9 ± 6.18). As for colistin doses, patients received 2.5–5.0 mg of CBA/kg of body weight daily, up to a maximum dose of 300 mg (10 million IU). However, the mean daily dose of colistin was 152.8 (5.1 million IU) ± 62.8 mg CBA (2.1 million IU) (range 60–300 mg CBA). The clinical cure rate in that study was 25%, although the authors did not perform a specific analysis for each type of microorganism (13). In another study of critically ill patients with serious infections caused by antimicrobial-resistant *P. aeruginosa*, the authors included 18 patients with pneumonia. Initial colistin doses were based on ideal body weight and estimated creatinine clearance, and colistin could be administered as monotherapy or adjunctive therapy. A total of 10 out of 18 (55.5%) patients showed favorable clinical outcomes (14), but there was no specific information on doses and therapeutic regimens in those patients diagnosed with pneumonia. Finally, in the study by Montero et al., mortality among patients diagnosed with pneumonia caused by XDR *P. aeruginosa* was 35% (15). CMS doses in that study ranged from 2.5 to 5 mg/kg/day with a median daily dose of 240 million IU (120–480 million IU), equivalent to 90 mg (45–180 mg). Importantly, in that study, 35 patients (28.9%) received intravenous and nebulized CMS, and 69.4% were treated with combination therapy. The heterogeneity of these studies (dosage regimen, different antibiotic combinations, use of nebulized colistin), together with the lack of specific information on the subgroup of patients with pneumonia, makes comparisons difficult. Even so, it can be stated that mortality in pneumonia caused by *P. aeruginosa* continues to be very high and gives cause for concern.

In our study, the patients who died were more seriously ill and achieved higher colistin plasma concentrations. While SOFA score as a predictor of mortality in patients with *P. aeruginosa* infections has been reported previously (10, 16, 17), the finding that higher colistin plasma concentrations are related to mortality is a new factor to take into consideration.

Recent population pharmacokinetic studies and consensus guidelines have suggested a specific target colistin plasma concentration of 2 mg/L (9, 18, 19). This target, however, is considered appropriate for the treatment of *P. aeruginosa* infections with an MIC of 1 mg/L and should, therefore, be adapted to local epidemiology. Given that the most common strain in our hospital has an MIC of 0.5 mg/L, an AUC/MIC of 30–60 mg·h/L would be a reasonable PK/PD target. This may help to explain why colistin plasma concentrations lower than those recommended in clinical guidelines and consensus documents were, rather unexpectedly, not associated with worse clinical outcomes.

Our findings are in line with a recent study whose authors studied the population pharmacokinetics of colistin and its association with time to death and found that higher plasma exposure to colistin was linked to an increased risk of mortality [HR (95% CI): 1.07 (1.03–1.12)] (10). Although there are some differences between the two studies in terms of patient and infection characteristics (our patients were older, had higher Charlson comorbidity index, lower SOFA scores, and all had HAP caused by XDR *P. aeruginosa*) and CMS dosage schedules (loading dose of 9 million IU of CMS, followed by 4.5 million IU

every 12 hours in the Kristofferson study vs a median CMS dose of 5.54 million IU/day in the present one), colistin plasma concentrations were similar: 0.44 (0.14–1.59) mg/L in samples drawn at 45 min after the start of infusion in the Kristofferson study vs 1.1 (0.56–1.75) mg/L in samples drawn at pre-dose in the present one. Despite the similarities in the findings of the two studies, these results should be interpreted with caution. It is difficult to explain *a priori* why high plasma exposure to colistin should be associated with 30-day all-cause mortality. It is well known that colistin plasma concentrations are related to colistin-associated nephrotoxicity, and our group and others have shown that nephrotoxicity is a risk factor for mortality (20–22). However, we were unable to demonstrate this association in the present study, probably due to the small sample size. With respect to clinical failure, we consider it important to point out that the only associated factors were the SOFA score and $C_{ss,avg}$. In a previous study conducted by our group on individuals with XDR *P. aeruginosa* infections, we observed the same results (17). Again, it is difficult to explain why individuals with high colistin exposure have poorer clinical cure rates.

A combined analysis of all these results suggests that an approach guided by colistin plasma levels is required to prevent overexposure and, consequently, colistin-associated nephrotoxicity in patients with HAP caused by *P. aeruginosa*. Nevertheless, there remains some concern about the therapeutic target of colistin and the role of intravenous CMS in the treatment of pneumonia. While colistin penetration into epithelial lining fluid (ELF) following intravenous administration has not been extensively studied, the results available to date suggest that colistin penetration into the lung is insufficient for the treatment of pneumonia (23–25). Furthermore, the proposed therapeutic plasma concentrations (2 mg/L) are difficult to achieve in patients with good renal function and overlap with nephrotoxic plasma concentrations in those with normal or poor renal function (9, 21, 26). These findings, taken together, support the idea that exposure to antibiotics at subtherapeutic concentrations at the site of infection may be responsible for the high mortality rate in patients with *P. aeruginosa* pneumonia and that this cannot be addressed by increasing exposures due to toxicodynamic concerns.

In this study, colistin-associated nephrotoxicity was detected in almost 40% of patients. However, most of the patients were classed as R and I (according to RIFLE criteria), suggesting low-level toxicity. These results are similar to those reported by our group in previous studies (16, 17, 21). Even though we cannot demonstrate a correlation between nephrotoxicity and mortality in the present study, we believe it is important to point out that more than one-third of patients showed some degree of renal injury, given the correlation between nephrotoxicity and poor clinical outcomes, which has been previously established (17, 21, 22).

The limitations of this study include the small sample size and heterogeneity of treatment regimens. Almost 60% of patients received nebulized colistin and 70% combination therapy. With respect to combination therapy, clinical trials have not been able to demonstrate its superiority over monotherapy (27). Data on inhaled colistin meanwhile are scarce. Ideally, a prospective comparative trial of nebulized vs intravenous colistin would be performed to assess the comparative effectiveness of these two strategies for XDR *P. aeruginosa* pneumonia. Until such a study is performed, our present study provides real-world clinical information on the use of colistin in this clinical setting.

In conclusion, colistin use in the setting of HAP caused by XDR *P. aeruginosa* is associated with a high mortality rate, and the risk of mortality seems to be higher in patients with higher colistin exposure. The finding that patients with colistin plasma concentrations > 1.5 mg/L showed a worse clinical outcome reinforces the idea that the PK/PD of colistin in this clinical setting is yet to be determined and that $C_{ss,avg}$ in plasma may not be a surrogate marker of ELF colistin concentrations. These findings, together with the high rate of colistin-associated nephrotoxicity, suggest that intravenous colistin should be regarded as a "last-line" agent and that safer molecules should be used when possible. We also suggest nebulized colistin as a valid option to consider in this special

situation, taking into account the PK/PD at the infection site, although more studies are needed before this practice can be strongly recommended.

## MATERIALS AND METHODS

### Study design and participants

This study was conducted in a tertiary care university hospital in Barcelona (Spain) from January 2010 to September 2018. Eligibility criteria were admitted patients (≥18 years) diagnosed with HAP caused by XDR *P. aeruginosa*, treated with intravenous CMS, either as monotherapy or in combination therapy, and had measured plasma concentrations of formed colistin. Exclusion criteria were <18 years old, pregnancy, breastfeeding during the study period, allergy to polymyxins, having a concomitant infection or an infection caused by colistin-resistant *P. aeruginosa*, and being on renal replacement therapy.

### Pneumonia diagnosis and treatment regimens

Diagnosis of pneumonia was based on new or progressive infiltrates on chest x-rays and at least two clinical findings suggestive of pneumonia, including cough, purulent sputum, fever or hypothermia (>38℃ or <36℃), and leukocytosis or leucopenia (>12,000/mL or <4,000 /mL). Microbiological diagnosis of sputum, endotracheal aspirate, or bronchoalveolar lavage culture with *P. aeruginosa* was valid at a diagnostic threshold of ≥$10^4$ colony-forming units per millimeter (28), or when blood cultures were positive. Intravenous antibiotics, including CMS, meropenem, ceftazidime, and amikacin, as well as nebulized CMS (when it was prescribed by the responsible physician) were administered according to local guidelines for 10–14 days (Table S1). The final treatment duration was decided by the treating physician. Standard CMS doses were 2 or 3 million IU (60–90 mg colistin base activity) every 8 hours in patients with normal renal function. In patients with estimated creatinine clearance <90 mL/min, the dosage was adjusted according to hospital protocol (29) (Table S1).

Patients treated with nebulized CMS (when it was administered), patients received a dosage of 2–3 million IU every 8 hours, which is equivalent to 100 mg of colistin base activity every 8 hours. The nebulization of CMS was conducted using a vibrating-mesh nebulizer with a maximum capacity of 6 mL (Aeroneb Pro, Aerogen, Galway, Ireland) or a jet nebulizer powered by an air/oxygen mixture (Kendall, Covidien, Dublin, Ireland). The administration of a dosage of 3 million IU/8 hours of CMS requires 6 mL of diluent, and the nebulized chamber was filled once and nebulized for 30 min. In patients undergoing mechanical ventilation and utilizing a vibrating-mesh nebulizer, it was seen that optimal efficiency was achieved by relocating the nebulizer to the inspiratory limb, specifically at a distance of 15 cm from the T-piece. This adjustment involved the removal of the humidifier. In patients undergoing assist-control mode ventilation and receiving sedation, ventilation was performed with a constant flow targeting an inspiration-to-expiration time ratio of 0.5 and respiratory rates of 12–14 cycles/min. Alternatively, in cases where patients did not get adequate sedation to facilitate modifications of the ventilator parameters, such adjustments were intentionally omitted to prevent excessive sedation.

### Patient data collection

Data collected from patients included demographic and anthropometric information, SOFA score (30), Charlson comorbidity index (31), baseline serum creatinine, and estimated glomerular filtration rate by MDRD-4 (32) at baseline, day 7, and end of treatment. The development of colistin-associated nephrotoxicity was also assessed and classified according to RIFLE criteria (20).

## Microbiological data

Identification and susceptibility testing of *P. aeruginosa* were performed by microdilution, using the Gram-negative breakpoint panel for non-fermenting Gram-negative bacteria on the MicroScan WalkAway system (Siemens Diagnostic Inc., CA, USA). The MIC of colistin was determined by microdilution using cation-adjusted Mueller-Hinton broth (MHB); the isolate was considered susceptible if the MIC was ≤2 mg/L, according to Clinical and Laboratory Standards Institute guidelines (33). When the MIC was not available (*n* = 18), the median MIC value (0.5 mg/L) of the most common *P. aeruginosa* strain in our center (ST175 clone) was imputed.

## Sampling and bioanalysis

Each dose of CMS was diluted in 100 mL of sterile saline and administered intravenously over 30 min. Blood samples were collected just before the next dose on day 3–4 of treatment, when it was assumed that a steady state had already been achieved (>4 half-lives considering a median half-life of 14 hours) (34). Concentrations of CMS and formed colistin in plasma were measured as reported previously (35, 36). The limits of quantification for colistin and CMS in plasma were 0.20 and 0.50 mg/L, respectively. Based on the almost flat plasma concentration-time profiles of formed colistin at steady state in critically ill patients (9, 10, 21), the measured colistin concentrations were considered as $C_{ss,avg}$. Since the most predictive PK/PD index for colistin is the ratio of area under the unbound (free) plasma concentration-time curve to minimum inhibitory concentration ($fAUC_{24}/MIC$) (36, 37), the $AUC_{24}/MIC$ was calculated for each patient. The results of murine models of lung infection caused by *P. aeruginosa* showed that an $AUC_{24}/MIC$ ratio of 60 was generally associated with an effect between stasis and 1-log kill in three strains of *P. aeruginosa* (11). However, based on pharmacokinetic/toxicodynamic studies, the International Guidelines for the optimal use of polymyxins recommended an $AUC_{24}/MIC$ of ~50 mg·h/L, equivalent to a target $C_{ss,avg}$ of ~2 mg/L for the total drug (26, 38). In light of these data, plus the fact that the $MIC_{50}$ of the most prevalent strain in our hospital is 0.5 mg/L, an $AUC_{24}/MIC$ in the range of 30–60 mg·h/L was deemed to be the appropriate exposure level. $AUC_{24}/MIC < 30$ mg·h/L was considered underexposure and $AUC_{24}/MIC \geq 60$ mg·h/L was considered overexposure. These values correspond to $C_{ss,avg}$ of 0.625–1.25 mg/L, $C_{ss} < 0.625$, and $C_{ss} \geq 1.25$ mg/L, respectively.

## Outcome evaluation

The primary endpoint was to assess the relationship between colistin plasma concentrations ($C_{ss,avg}$) and 30-day all-cause mortality. Secondary endpoints included the possible association of $C_{ss,avg}$ with clinical response, classified as cure (resolution of symptoms/signs and antibiotic-free) or failure (persistence or progression of signs/symptoms), or with the development of colistin-associated nephrotoxicity. Microbiological response was considered positive if *P. aeruginosa* was not isolated in repeat samples during or after the course of CMS therapy (eradication). Colonization was considered to be present when *P. aeruginosa* persisted in repeated sputum or bronchoalveolar lavage samples taken when the patient was stable and there was no clinical evidence of infection. Finally, superinfection was defined as a second infection with a different bacterium superimposed on the first one.

## Statistical analysis

Descriptive statistics were reported as median [interquartile range; (range)], numbers and relative frequencies, or mean ± standard deviation. Figures are presented as box (median, IQR) and whisker (10th and 90th percentiles) plots. Dichotomous variables were compared using the $X^2$ or Fisher's exact test, as appropriate. Continuous data with normal distribution were compared using the Mann-Whitney *U*-test. The outcome assessed in the exposure-response analysis was time to death, with censoring at 30 days after the start of colistin. The Cox proportional hazards regression model was used to

perform multivariate analyses of survival and 30-day all-cause mortality, and results were reported as HR and 95% confidence interval. Variables with $P$ value $\leq 0.2$ in univariate analysis and clinically relevant variables were included in the multivariate models. $C_{ss,avg}$ was included in the model as both a continuous and a categorical variable based on the degree of exposure to colistin (optimal, underexposure, and overexposure). The cutoff point that best-predicted mortality was selected by the maximally selected standardized log-rank statistic. All $P$-values were two-tailed and statistical significance was <0.05. Statistical analyses were performed using STATA 15.1. The STROBE guidelines were used to ensure adequate reporting of the study (Table S2).

## ACKNOWLEDGMENTS

We thank Mr. Xavier Duran as a statistical advisor and Janet Dawson for English editing.

S.L. received support from the Instituto de Salud Carlos III (grant INT21/ 00009, Contrato para la Intensificación de la Actividad Investigadora del Sistema Nacional de Salud).

This study was carried out as part of our Therapeutic Drug Monitoring Program.

## AUTHOR AFFILIATIONS

[1]Infectious Diseases Service, Hospital del Mar, Barcelona, Spain

[2]Infectious Pathology and Antimicrobials Research Group (IPAR), Institut Hospital del Mar d'Investigacions Mèdiques (IMIM), Barcelona, Spain

[3]Department of Medicine and Life Sciences, Universitat Pompeu Fabra (UPF), Barcelona, Spain

[4]CIBER of Infectious Diseases (CIBERINFEC CB21/13/00002 and CB21/13/00099), Institute of Health Carlos III, Madrid, Spain

[5]Pharmacy Service, Hospital del Mar, Barcelona, Spain

[6]Infectious Pathology and Antimicrobials Research Group (IPAR), Institut Hospital del Mar d'Investigacions Mèdiques (IMIM), Barcelona, Spain

[7]Infection Program and Department of Microbiology, Monash Biomedicine Discovery Institute, Monash University, Melbourne, Victoria, Australia

[8]Department of Anesthesiology and Surgical Intensive Care, Infectious Pathology and Antimicrobials Research Group, Institut Hospital del Mar d'Investigacions Mèdiques, Barcelona, Spain

[9]Microbiology Service, Laboratori de Referència de Catalunya, Barcelona, Spain

[10]Analytical Department, Laboratori de Referència de Catalunya, Barcelona, Spain

## AUTHOR ORCIDs

Luisa Sorlí http://orcid.org/0000-0001-9562-514X
Sonia Luque http://orcid.org/0000-0002-3215-320X
Jian Li http://orcid.org/0000-0001-7953-8230
Silvia Gómez-Zorrilla https://orcid.org/0000-0002-5987-068X

## FUNDING

| Funder | Grant(s) | Author(s) |
| --- | --- | --- |
| MEC | Instituto de Salud Carlos III (ISCIII) | INT21/ 00009 | Sonia Luque |

## AUTHOR CONTRIBUTIONS

Luisa Sorlí, Conceptualization, Data curation, Formal analysis, Investigation, Methodology, Project administration, Supervision, Validation, Writing – original draft, Writing – review and editing | Sonia Luque, Conceptualization, Data curation, Formal analysis, Investigation, Methodology, Project administration, Supervision, Writing – original draft, Writing – review and editing | Jian Li, Supervision, Writing – review and editing |

Adela Benítez-Cano, Data curation, Writing – review and editing | Nuria Prim, Investigation | Victoria Vega, Investigation | Joan Gómez-Junyent, Investigation, Methodology | Inmaculada López-Montesinos, Investigation | Silvia Gómez-Zorrilla, Investigation, Methodology | M. Milagro Montero, Investigation | Santiago Grau, Investigation | Juan Pablo Horcajada, Supervision.

## ETHICS APPROVAL

The study was conducted according to the principles set out in the Declaration of Helsinki, and ethical approval was obtained from the local ethics committee (CEIm-Parc de Salut MAR).

## ADDITIONAL FILES

The following material is available online.

### Supplemental Material

**Tables S1-S4 and Figures S1-S3 (Spectrum02967-23-S0001.docx).** Antibiotic doses according to local guidelines, STROBE document, $C_{ss,avg}$ related to estimated GFR, Multivariate analysis of mortality and clinical cure including Css as categorical variable and Kaplan-Meier also with $C_{ss,avg}$ as categorical variable. Finally, a figure with Css avg cutoff point according to the maximally selected standardized log-rank statistic.

### Open Peer Review

**PEER REVIEW HISTORY (review-history.pdf).** An accounting of the reviewer comments and feedback.

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
