## [Reviewer comments · Microbiology Spectrum]

Microbiology Spectrum

Colistin Plasma Concentrations Are Not Associated with a Better Clinical Outcomes in Patients with Pneumonia caused by Extremely Drug-Resistant *Pseudomonas aeruginosa*

Luisa Sorlí, Sonia Luque, Jian Li, Adela Benítez-Cano, XENIA FERNÁNDEZ, Nuria Prim, Victoria Vega, Joan Gómez-Junyent, Inmaculada López Montesinos, Silvia Gómez-Zorrilla, María Milagro Montero, Santiago Grau, and Juan Pablo Horcajada

Corresponding Author(s): Luisa Sorlí, Institut Hospital del Mar d'Investigacions Mediques

Review Timeline:

Submission Date:	July 27, 2023
Editorial Decision:	September 5, 2023
Revision Received:	September 25, 2023
Accepted:	October 9, 2023

Editor: Fikri Avci

Reviewer(s): Disclosure of reviewer identity is with reference to reviewer comments included in decision letter(s). The following individuals involved in review of your submission have agreed to reveal their identity: Vijay K Singh (Reviewer #1)

Transaction Report:

DOI: <https://doi.org/10.1128/spectrum.02967-23>

September 5, 2023

Dr. Luisa Sorlí
Institut Hospital del Mar d'Investigacions Mediques
Infectious Diseases
Passeig Marítim 25-29
Barcelona, Barcelona 08003
Spain

Re: Spectrum02967-23 (An 8-Year Experience of Therapeutic Drug Monitoring of Colistin in Patients with Pneumonia Caused by Extremely Drug-Resistant *Pseudomonas aeruginosa*)

Dear Dr. Sorlí:

Thank you for submitting your manuscript to Microbiology Spectrum. We have completed our review of your manuscript, and I am pleased to inform you that, in principle, we expect to accept it for publication. However, acceptance will not be final until you have addressed the minor reviewer comments.

Link Not Available

Sincerely,

Fikri Avci

Journals Department
Reviewer comments:

Reviewer #1 (Comments for the Author):

The study by Sorli et al. is very important and interesting. The study is well-designed, and the conclusion is sound. I have very minor comments.

1. The title Should be modified.

2. A short description of figure 1 should be added.

Reviewer #2 (Comments for the Author):

In their manuscript, "An 8-Year Experience of Therapeutic Drug Monitoring of Colistin in Patients with Pneumonia Caused by Extremely Drug-Resistant *Pseudomonas aeruginosa*" Sorli and colleagues are investigating the relationship between colistin plasma concentration and the clinical outcome in patients with HAP caused by XDR *Pseudomonas aeruginosa*. The manuscript is one of a series of many published by the same group investigating the outcomes of using colistin in combating XDR-PA. In their findings and conclusions, the authors found that higher colistin plasma concentration was associated with a worse clinical outcome. And they suggested that nebulized colistin holds promise as a valuable treatment option for nosocomial pneumonia.

Comments:

Line 103-109: Appropriate reference(s) need to be included for these statements "New information on the pharmacokinetics of colistinhas been recently published" and "Based on PK/PD from mouse data"

For the study design and treatment regimen, I am confused by how CMS was administered. It's not clear to me whether the patient cohort of choice were those received intravenous CMS (alone or in combination therapy). Then in lines 133-134, the authors mention nebulization. Was CMS nebulized alone or in combination with other antibiotics? And was CMS administered separately or concurrently as IV and inhaled treatment (Table 1)?

- In the sampling and the bioanalysis, there's no description on how the CMS was nebulized.

Line 270: "A high mortality rate (30.7%) was observed in our series" Can the authors be more specific? Are they referring to previous work done by their group? What are these specific references?

Line 343: Is that the correct reference (Markou et al)?

Figure S1 need to be in better quality.

Staff Comments:

Preparing Revision Guidelines

Please return the manuscript within 60 days; if you cannot complete the modification within this time period, please contact me. If you do not wish to modify the manuscript and prefer to submit it to another journal, please notify me of your decision immediately so that the manuscript may be formally withdrawn from consideration by Microbiology Spectrum.

“RESPONSE TO REVIEWERS”

Manuscript ID: Spectrum02967-23

Title: An 8-Year Experience of Therapeutic Drug Monitoring of Colistin in Patients with Pneumonia Caused by Extremely Drug-Resistant *Pseudomonas aeruginosa*

Authors: Luisa Sorli, Sonia Luque, Jian Li, Adela Benítez-Cano, Xenia Fernández, NuriaPrim, Victoria Vega, Joan Gómez-Junyent, Inmaculada López-Montesinos, Silvia Gómez-Zorrilla, M. Milagro Montero, Santiago Grau, Juan Pablo Horcajada

Dear Professor Fikri Avci:

Thanks for the opportunity to review our manuscript. We have carefully reviewed it and here are the answers to the reviewers' comments point-by-point:

RESPONSE TO REVIEWERS POINT-BY-POINT

Reviewer 1:

Comments to the Author:

The study by Sorli et al. is very important and interesting. The study is well-designed, and the conclusion is sound. I have very minor comments.

1. The title Should be modified.

According to the reviewer's recommendation, the title has been updated.

We believe that the new title places greater emphasis on the work's results. If it does not satisfy the reviewer, we are open to any suggestions.

2. A short description of figure 1 should be added.

Thank you for your comment. As suggested by the reviewer, a concise description of the Figure 1 has been provided.

Reviewer 2:

Comments to the Author:

In their manuscript, "An 8-Year Experience of Therapeutic Drug Monitoring of Colistin in Patients with Pneumonia Caused by Extremely Drug-Resistant Pseudomonas aeruginosa" Sorli and colleagues are investigating the relationship between colistin plasma concentration and the clinical outcome in patients with HAP caused by XDR Pseudomonas aeruginosa. The manuscript is one of a series of many published by the same group investigating the outcomes of using colistin in combating XDR-PA. In their findings and conclusions, the authors found that higher colistin plasma concentration was associated with a worse clinical outcome. And they suggested that nebulized colistin holds promise as a valuable treatment option for nosocomial pneumonia.

Comments:

- 1. Line 103-109: Appropriate reference(s) need to be included for these statements" New on information on the pharmacokinetics of colistinhas been recently published" and "Based on PK/PD from mouse data"**

*Thank you for the comment. As suggested by the reviewer, the references have been inserted. References nine and ten correspond to clinical pharmacokinetic investigations, while eleven corresponds to a murine pharmacokinetic model. **Lines 89-91.***

- 2. For the study design and treatment regimen, I am confused by how CMS was administered. It's not clear to me whether the patient cohort of choice were those received intravenous CMS (alone or in combination therapy). Then in lines 133-134,**

the authors mention nebulization. Was CMS nebulized alone or in combination with other antibiotics? And was CMS administered separately or concurrently as IV and inhaled treatment (Table 1)?

According to the inclusion criteria, the study is focused on patients treated with intravenous CMS. Nonetheless, some of the included patients also received nebulized CMS. In addition, patients could receive monotherapy or combination therapy with CMS at the discretion of their treating physician. According to the reviewer's recommendation, the terminology has been revised and highlighted in the text. Lines 289-290 and 302-305.

3. In the sampling and the bioanalysis, there's no description on how the CMS was nebulized.

Thanks for the suggestion. We have added a new paragraph describing the nebulization of CMS. Lines 311-327

3. Line 270: "A high mortality rate (30.7%) was observed in our series" Can the authors be more specific? Are they referring to previous work done by their group?

What are these specific references?

We appreciate your comments. In fact, this mortality rate is from the present study. We have revised the text accordingly. Line 156

4. Line 343: Is that the correct reference (Markou et al)?

Thank you very much. This is an error. This citation is from one of our previous works, so we have updated the manuscript accordingly. Line 235.

5. Figure S1 need to be in better quality.

Thank you for your comment. The graph contains a great deal of potentially superfluous information. We have modified the graph to improve the quality and to make it more understandable.

October 9, 2023

Dr. Luisa Sorlí
Institut Hospital del Mar d'Investigacions Mediques
Infectious Diseases
Passeig Marítim 25-29
Barcelona, Barcelona 08003
Spain

Re: Spectrum02967-23R1 (Colistin Plasma Concentrations Are Not Associated with a Better Clinical Outcomes in Patients with Pneumonia caused by Extremely Drug-Resistant *Pseudomonas aeruginosa*)

Dear Dr. Sorlí:

Congratulations, your manuscript has been accepted, and I am forwarding it to the ASM Journals Department for publication. You will be notified when your proofs are ready to be viewed.

Sincerely,

Fikri Avci
Editor, Microbiology Spectrum
